# CUSTOMIZED PROCEDURE PLANNING IN INSTRUCTIONAL VIDEOS

## ABSTRACT

Generating customized procedures for task planning in instructional videos poses a unique challenge for vision-language models. In this paper, we introduce Customized Procedure Planning in Instructional Videos, a novel task that focuses on generating a sequence of detailed action steps for task completion based on user requirements and the task's initial visual state. Existing methods often neglect customization and user directions, limiting their real-world applicability. The absence of instructional video datasets with step-level state and video-specific action plan annotations has hindered progress in this domain. To address these challenges, we introduce the Customized Procedure Planner (CPP) framework, a causal, open-vocabulary model that leverages a LlaVA-based approach to predict procedural plans based on a task's initial visual state and user directions. To overcome the data limitation, we employ a weakly-supervised approach, using the strong vision-language model GEMINI and the large language model (LLM) GPT-4 to create detailed video-specific action plans from the benchmark instructional video datasets COIN and CrossTask, producing pseudo-labels for training. Discussing the limitations of the existing procedure planning evaluation metrics in an open-vocabulary setting, we propose novel automatic LLM-based metrics with few-shot in-context learning to evaluate the customization and planning capabilities of our model, setting a strong baseline. Additionally, we implement an LLM-based objective function to enhance model training for improved customization. Extensive experiments, including human evaluations, demonstrate the effectiveness of our approach, establishing a strong baseline for future research in customized procedure planning.

## 1 INTRODUCTION

Procedure planning in instructional videos (PPIV) involves generating a sequence of action steps, to transform an initial visual observation of a task into its completion (Chang et al., 2020; Bi et al., 2021a; Sun et al., 2022; Zhao et al., 2022; Wang et al., 2023a;b; Li et al., 2023; Niu et al., 2024; Zare et al., 2024; Nagasinghe et al., 2024). Autonomous agents capable of performing this task can assist humans in efficiently completing complex, goal-oriented tasks and procedures in daily life. While humans intuitively understand the steps and reasoning needed to accomplish such tasks, machines face considerable challenges in replicating this ability. To overcome this gap, an autonomous agent requires a deep understanding of instructional procedures, their unique characteristics, related objects, the various states involved, and the transformations brought about by actions. This understanding is essential for generating a plausible, executable plan that leads to successful task completion.

Despite considerable progress in recent studies, various obstacles still restrict its practical applications in the real world. Recent works on procedure planning in instructional videos have largely overlooked the importance of customization and user-specific directions. Most existing approaches rely on initial and final visual observations of a task, resulting in a non-causal formulation (Chang et al., 2020; Bi et al., 2021a; Sun et al., 2022; Zhao et al., 2022; Wang et al., 2023a;b; Li et al., 2023; Niu et al., 2024; Zare et al., 2024), which limits their applicability in real-life scenarios. This reliance on visual information alone introduces a semantic gap, particularly in representing intermediate action steps that may depend on user-specific conditions but are not captured by the visual inputs. Consequently, the generated action plans often lack informativeness, producing generic se-

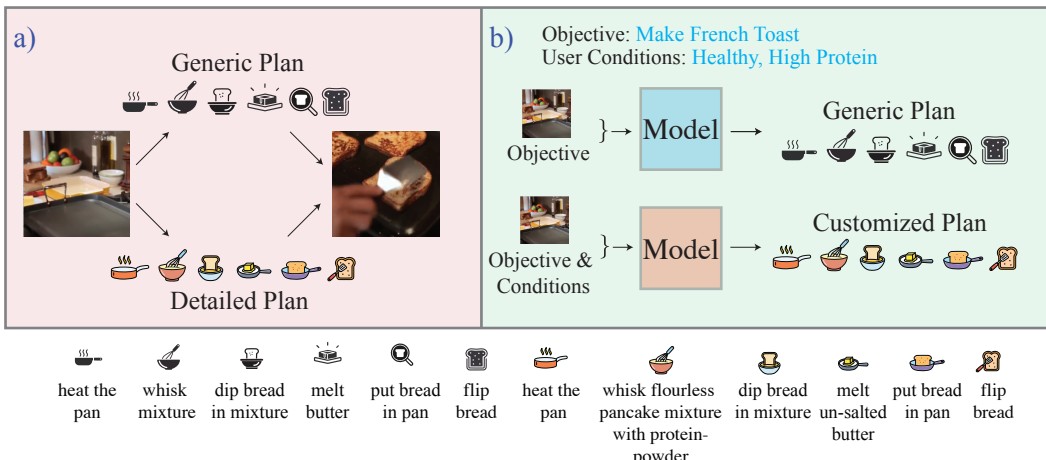

Figure 1: **(a)** Illustration of the semantic gap in procedure planning, where the initial and final visual states do not distinguish between a generic and detailed plan, resulting in ambiguity. **(b)** Comparison of two settings: a model that integrates user-specified keyword conditions produces a customized and informative instructional plan, while a model relying solely on task objectives lacks essential detail. The bottom model demonstrates the practical setting of customized procedure planning in instructional videos.

quences blind to user-specific needs. This issue is illustrated in Fig. 1a, where the initial and final visual states do not distinguish between a generic and a more detailed plan, resulting in ambiguity and reinforcing the semantic gap.

Some models, such as (Wang et al., 2023a), which incorporate textual inputs, have made progress in bridging this semantic gap between visual observations and intermediate steps. However, they still fall short by conditioning planning solely on task-related textual information inferred from the observed states, without generating action steps tailored to user-specific directions or conditions necessary to complete a task from its current state.

A model that fully addresses this limitation must go beyond simple visual inputs. It should be capable of processing both the current visual state of the task and user-specific requirements provided in textual form. This would allow the model to generate a more tailored plan, transforming the task toward completion in a way that aligns with both the visual state and the user's directions. Fig. 1b highlights the contrast between the two approaches: a model that incorporates user-specified needs, such as keyword conditions, can produce a more customized, detailed, and informative instructional plan with customized steps. This stands in contrast to a model that relies solely on task objectives, showcasing the practicality and relevance of customized procedure planning.

Addressing this need cannot be adequately captured within the conventional closed-vocabulary setting under which this problem has been studied (Chang et al., 2020; Bi et al., 2021a; Sun et al., 2022; Zhao et al., 2022; Wang et al., 2023a;b; Li et al., 2023; Niu et al., 2024; Zare et al., 2024), as it restricts plan prediction to predefined action labels. While recent works, such as Wu et al. (2024), have made progress in expanding the problem of PPIV to an open-vocabulary setting, the challenge of generating detailed, user-specific action plans remains unresolved.

A key challenge in extending PPIV to address user-specific needs has been the lack of suitable datasets for training. To train such a model, a large dataset of instructional videos is required, along with their corresponding detailed instructional plans, annotated with time-stamped procedural states. These detailed plans must be tailored to the specific characteristics of each video, which distinguish an instructional video from more generic ones, addressing unique user requirements. However, obtaining such annotations is both expensive and time-consuming. Existing benchmark datasets for this task, such as CrossTask and Coin (Zhukov et al., 2019; Tang et al., 2019), provide step-level annotations of procedural states and generic plans, but they lack the detailed instructional plans and video-specific characteristics that make each instructional plans informative and unique in terms of user demands.

We tackle these challenges, by introducing the setting of Customized Procedure Planning in Instructional Videos (CPPIV) and proposing the Customized Procedure Planner (CPP) framework as a solution for this problem. We implement CPP as a LlaVa-based (Liu et al., 2023; 2024a;b) model, fine-tuned to generate detailed, open-vocabulary instructional plans for task completion, starting from an initial visual state and customized based on user-specified keywords.

To overcome dataset limitations in training CPP, we adopt a weakly supervised approach. First, we leverage the powerful vision-language model, GEMINI (Team et al., 2023), to extract video-specific, task-related keywords and generate descriptions that explain how these keywords are relevant to the video's action plan. This is applied to the CrossTask and COIN datasets. Using this customized information—key elements that differentiate the instructional content of each video from generic task plans—we conditionally generate a customized, video-specific instructional plan. To achieve this, we employ the strong LLM, GPT-4o (OpenAI, 2023), to adapt the generic instructional ground truth plan for each video based on the extracted keywords. These customized plans serve as pseudo-labels for training the CPP model. Additionally, during training, GPT-4o is integrated into the objective function to further enhance the model's ability to produce customized instructional plans.

Extending Procedure Planning in Instructional Videos to an open-vocabulary setting presents challenges for traditional evaluation metrics, which rely on pre-defined, closed-vocabulary action step labels and fail to generalize effectively. To overcome this, we draw on recent works (Liang et al., 2023; Zhu et al., 2023; Wang et al., 2024; Huang et al., 2024), and introduce a novel LLM-based approach—referred to as automatic metrics—to assess the quality of both planning and customization in detailed, varied, open-vocabulary plans. We evaluate our model on two widely used instructional video datasets, CrossTask and COIN. Additionally, we validate our model's performance by testing it on human-annotated customized plans from both datasets. Our model outperforms the state-of-the-art (SoA) and establishes a strong baseline for the setting of customized procedure planning.

Our main contributions are:

– We emphasize the need for a more practical formulation of procedure planning in instructional videos that considers user directions and specific requirements and introduce the novel setting of customized procedure planning in instructional videos, aimed at generating instructional plans that cater to user task-specific needs rather than relying solely on generic task completion.

– We propose the Customized Procedure Planner framework, which generates open-vocabulary instructional plans tailored to user-specified condition keywords, facilitating the transformation of initial visual states into task completion.

– We propose a weakly supervised training approach that addresses the lack of customization annotations for CPPIV model training, allowing customized planning to be learned from unannotated videos.

– We extend conventional procedure planning metrics to encompass open-vocabulary, varied, and detailed instructional plans, enabling a comprehensive assessment of planning and customization performance for predicted plans.

## 2 RELATED WORKS

### 2.1 PROCEDURE PLANNING

Procedure planning from instructional videos involves generating effective task completion plans. Earlier works employed a two-branch architecture, sequentially predicting actions and states with recursive models (Jain & Medsker, 1999; Vaswani et al., 2017) to capture state transitions. More recent methods, such as Zhao et al. (2022); Wang et al. (2023b), generate plans using a single-branch architecture that directly decodes actions, minimizing prediction error propagation. However, these approaches rely solely on visual observations of the initial and final states, resulting in a non-causal formulation that lacks adaptability to user-specific tasks and needs. Our work introduces CPP, a novel one-branch prediction framework that generates a detailed sequence of actions based on both the initial visual state and user-defined conditions, addressing this limitation in the existing literature.

## 2.2 CONDITIONAL VISION-LANGUAGE MODELS FOR SEQUENCE GENERATION

The problem of customized procedure planning can be framed as Conditional Vision-Language Models for Sequence Generation. In this approach, the model generates an output sequence by conditioning on both visual input (i.e., the current visual state) and textual input (i.e., task and user requirements). To address the CPPIV challenge, we utilize the Conditional Vision-Language Models framework, leveraging models such as LLaVA (Liu et al., 2024a; 2023; 2024b) and GPT-4o (OpenAI, 2023).

## 2.3 AUTOMATIC-METRICS

Due to the lack of comprehensive benchmarks and metrics in previous literature, judging and evaluating open-ended LLM results can be burdensome. In the training process of LLMs themselves, the accurate evaluation of open-ended output is essential. There is a growing trend to use LLMs to perform instruction fine-tuning on other LLMs with Huang et al. (2024) suggesting that a fine-tuned judge model can achieve high performance on in-domain data. Other recent literature (Zhu et al., 2023) suggests that the use of LLMs as a judge model is a powerful and robust method to create scalable evaluations in an open-ended framework. These judge models (Wang et al., 2024) can accurately answer questions related to judging answer pairs, explaining judgements, grading single answers and can even extend these capabilities to multimodal answers. In this work, we face a similar obstacle in the form of a lack of standardized benchmarks for the evaluation of open-vocabulary plan sequences. As such, we build on these judge model techniques to create customized scoring metrics for these output sequences.

## 3 TECHNICAL APPROACH

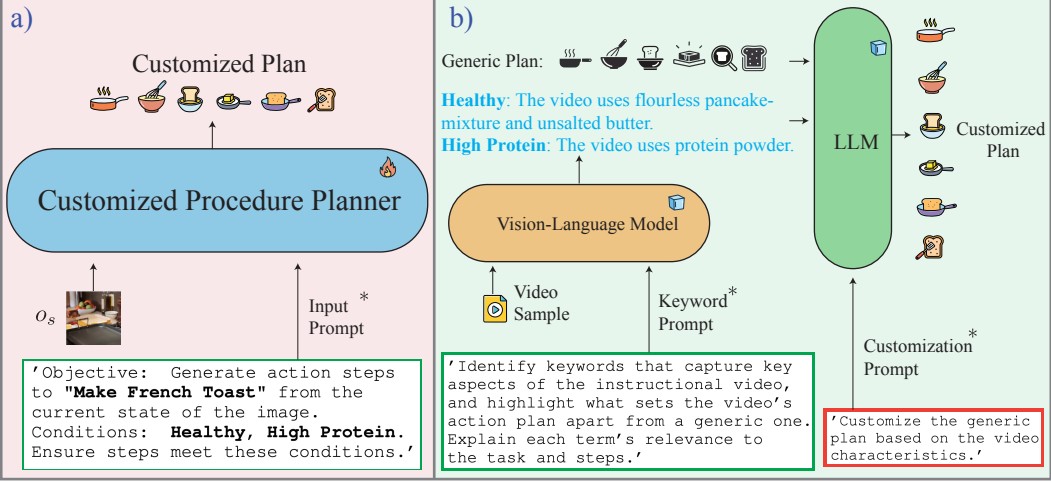

Figure 2: Overview of the Customized Procedure Planner (CPP) framework and data collection pipeline. (a) The CPP employs a vision-language model that takes a prompt with the task objective, user-defined conditions, and the current visual state $o_s$ to generate customized action steps. (b) The pipeline extracts task-specific keywords from the PPIV datasets using a vision-language model (VLM), which are then combined with a human-annotated generic plan to create pseudo labels for training customized instructional video datasets with the aid of a large language model (LLM). * Refer to prompts 1, 2, & 3 for the complete text.

In this section, we introduce our proposed framework, the Customized Procedure Planner, designed for customized procedure planning in instructional videos. We also explore the weakly supervised learning approach employed to train the CPP in the absence of datasets containing customization annotations.

## 3.1 SETTING: CUSTOMIZED PROCEDURE PLANNING

We define the novel setting of customized procedure planning in instructional videos as follows: Given an initial visual observation $o_s$, a task objective Task, and a sequence of user-specified customization keywords Keywords $= \{k_1, k_2, \ldots, k_K\}$, the model generates a plan $p = \{a_1, a_2, \ldots, a_T\}$, where $T$ represents the plan's length (i.e., the action horizon) and $a_i$ (for $1 \leq i \leq T$) is the detailed customized text for the $i$-th action step. This plan should effectively transform $o_s$ into the task objective while satisfying the specified customization conditions outlined by Keywords (see the bottom scenario in Fig. 1b).

## 3.2 MODEL: CUSTOMIZED PROCEDURE PLANNER

To implement the Customized Procedure Planner (CPP), we employ a vision-language model built on LLaVa. We experiment with LLaVa-1.5 (Liu et al., 2024a) and LLaVa-NeXT (Liu et al., 2024b) as the backbone of our framework, fine-tuning these models with pseudo-customized labels, as described in section 3.3. The operation of CPP is illustrated in Fig. 2a. The model takes as input $o_s$, a prompt containing the task objective Task, and user-defined conditions Keywords, and generates a sequence of customized action steps $p$. The zero-shot input prompt structure is shown in Prompt 1.

> Prompt 1: 'Objective: Compose a detailed sequence of action
> steps, in order, to complete the task "{**Task**}" depicted in
> the image, starting from its current state. Conditions:
> {**Keywords**}. Instructions: Ensure that the steps align
> with the specified conditions and lead to successful task
> completion.'

## 3.3 TRAINING

**Customizing Instructional Datasets.** Customized Procedure Planning suffers from a lack of sufficient datasets for training. We overcome this limitation by leveraging recent advancements in vision-language models (VLMs) and the capabilities of large language models (LLMs). As shown in Fig. 2b, our novel pipeline collects customizations from the PPIV datasets to build customized instructional video datasets, to use as pseudo labels for training. First, we employ the off-the-shelf vision-language model, GEMINI-1.5-Flash, to extract customization terms for each video sample in the datasets. These keywords are designed to be task-specific and tailored to the video's unique characteristics, as outlined in Prompt 2, which we implement along with a one-shot example response.

> Prompt 2: 'You will be provided with an instructional video
> that demonstrates a task through a series of ordered action
> steps (i.e., an instructional plan).
> Your response should identify up to 3 keywords for the video
> that are directly related to both the task and the action
> steps. These keywords should emphasize what distinguishes
> the video's instructional plan from a generic plan on the
> same task. For each term, provide a brief explanation of
> its relevance to the video, the task, and the action steps
> in one sentence.'

Next, we process the extracted keywords and their descriptions of how they relate to the video's instructional plan, alongside the corresponding human-annotated generic plan for the video. Using the GPT-4o LLM, guided by Prompt 3 and a one-shot example response, we generate a customized plan for each video (i.e., pseudo labels for training).

> Prompt 3: 'Compose a customized plan for an instructional
> video, based on the task and the video characteristics. The
> video includes a sequence of action steps, action-plan, in
> order. Format the response in one line. Your response
> should map each action step from the action-plan to a
> corresponding tailored customized step, maintaining the
> sequence order, in the format "'action step': tailored

```
        step", separated by commas.  If you need to include an
        additional step, use the term "added step".'
```

**Weak supervision.** With the generated pseudo-labels, we train the Customized Procedure Planner (CPP) using a cross-entropy loss function (Liu et al., 2024a;b). To improve the model's customization, we further incorporate the large language model (LLM) GPT-4o during training. GPT-4o is tasked with selecting of the best related plan to the Keywords, between two plans, $a$ and $b$ — one being the model's prediction and the other the pseudo-label, with the positions of $a$ and $b$ randomized. GPT-4o returns the error rate of its prediction, which is used to modify the overall batch loss.

The error rate is computed over the entire batch. For each sample in the batch, if GPT-4o's selection matches the pseudo-label, the sample accuracy is 1; otherwise, the accuracy is 0. The batch accuracy is the average accuracy across all samples in the batch, and the batch error rate is the complement of this accuracy, given by:

$$\text{Acc}_{\text{batch}} = \frac{1}{N} \sum_{i=1}^{N} \mathbb{1}(\hat{y}_i = y_i), \quad \hat{y}_i, y_i \in \{a, b\} \tag{1}$$

where: $N$ is the batch size, $\hat{y}_i$ is the LLM judge's selected plan for the $i$-th sample, $y_i$ is the corresponding pseudo-label for the $i$-th sample, $\mathbb{1}(\hat{y}_i = y_i)$ is an indicator function that equals 1 if $\hat{y}_i = y_i$, and 0 otherwise.

The batch error rate is calculated as:

$$\text{Error}_{\text{batch}} = 1 - \text{Acc}_{\text{batch}} \tag{2}$$

This error rate is scaled by a set positive factor $\lambda$ and added to the cross-entropy loss to adjust the training process, as described by the following equation:

$$\mathcal{L}_{\text{batch}} = \mathcal{L}_{\text{CE}} + \lambda \cdot \text{Error}_{\text{batch}} \tag{3}$$

Where $\mathcal{L}_{\text{CE}}$ is the cross-entropy loss between the predictions and the pseudo-labels, and $\lambda$ is a learnable scaling factor that controls the impact of the error rate on the batch loss.

## 4 EXPERIMENTS

We conduct experiments on two benchmark datasets, using novel evaluation metrics to validate the effectiveness of our proposed model, and further support our results through human evaluation.

### 4.1 DATASETS

We evaluate our methodology using two instructional video datasets: CrossTask (Zhukov et al., 2019) and COIN (Tang et al., 2019). The CrossTask dataset includes videos across 18 topics, such as "Make French Toast," with an average of 7.6 actions per video. These topics are split into 18 primary and 65 related events. In our study, we focus on the primary subset, which provides precise timestamps for each action, enabling a clear sequence of instructional steps and encompassing 2,750 videos. The COIN dataset contains 11,827 videos covering 778 distinct actions, with an average of 3.6 actions per video. Following recent works (Wang et al., 2023b; Bi et al., 2021b; Chang et al., 2020; Zhao et al., 2022), we create training and testing splits with a 70/30 ratio. To further enrich the datasets, we apply a moving window approach to organize videos into plans with varying action horizons. Starting from the $i$-th action, the window extends until the plan is complete (i.e., $T = |p| - i$).

Next, we apply the pseudo-label generation pipeline, as detailed in section 3.3 and Fig. 2b, to obtain customized plans for each dataset. This process leads to a more diverse set of action plans across the datasets. Fig. 3 illustrates the expansion of vocabulary in the action plans through word clouds, comparing the generic plans with the added vocabulary for four sample tasks. This emphasizes the open-vocabulary setting and the degree of customization achieved.

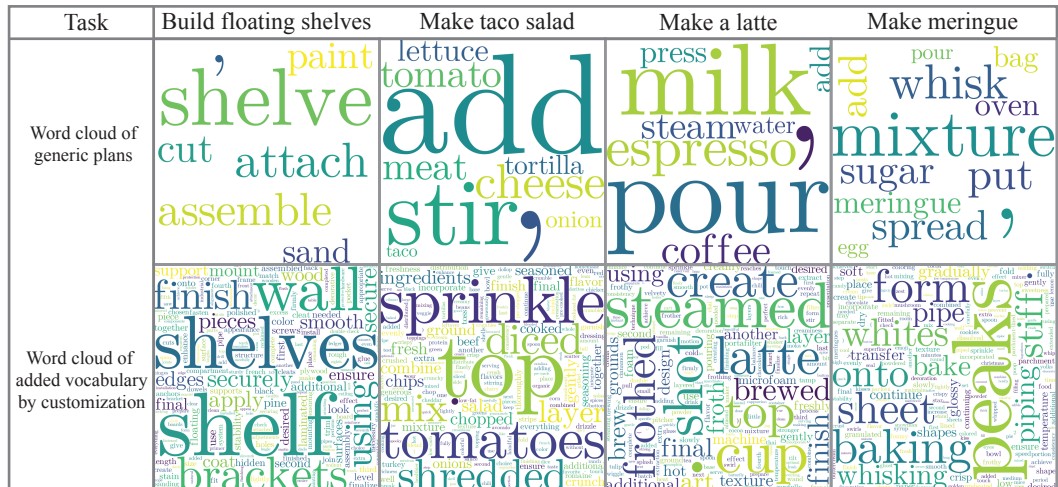

Figure 3: Expansion of vocabulary in action plans as the result of customization pipeline. The word clouds compare generic plans (top) with the added vocabulary (bottom) for four sample tasks, showcasing the open-vocabulary setting and customization on the CrossTask dataset. Stop-words are excluded (Bird et al., 2009) in the visualization.

## 4.2 METRICS

The performance of PPIV models is typically assessed using three standard metrics (Chang et al., 2020; Zhao et al., 2022; Sun et al., 2022; Bi et al., 2021a; Wang et al., 2023b): 1) Mean Intersection over Union (mIoU) evaluates the overlap between predicted and ground truth action sequences, defined as $\frac{|a_t \cap \hat{a}_t|}{|a_t \cup \hat{a}_t|}$. This metric indicates whether the model identifies the correct steps but does not account for action order or repetitions. 2) Mean Accuracy (mAcc) measures the alignment of actions at each step, taking into account the order and repetitions of actions. And 3) Success Rate (SR), the strictest metric, which considers a plan successful only if it precisely matches the ground truth.

However, all these metrics rely on action labels in both predicted and ground truth sequences, restricting the PPIV setting to a closed-vocabulary framework. This limitation impedes the evaluation of more practical open-vocabulary and varied plan sequences. In this study, we introduce four novel evaluation metrics that retain the essence of the conventional metrics while accommodating this new setting.

**Automatic Metrics.** This study has to quantify the performance of proposed plans in two different dimensions: Planning quality and customization quality. As mentioned, the nature of an open-vocabulary framework necessitates a novel approach to standard planning metrics found in previous literature. To this end, we combine Few-Shot-In-Context Learning and LLMs to create automatic metrics that are able to robustly score plans based on the two dimensions.

With regards to the quality of planning, we use Few-Shot-In-Context Learning combined with GPT-4o to create two types of sequence mappings from the predicted sequence to the closed-vocabulary generic ground-truth sequence. The first mapping is order mapping. Order mapping is a sequential process that iterates over the ground truth sequence. For each step $s_n = a_{n,GT}^{\text{Generic}}$ in the sequence, it tries to map it to a corresponding step $p_m = a_{m,Pred}^{\text{Customized}}$ in the predicted sequence. If unable to find a valid corresponding step $p_m$, it denotes the step $s_n$ as missing. The mapping proceeds with the next ground truth step $s_{n+1}$, which is only able to map to predicted sequence steps $p_{m+1}, ..., p_M$. This approach preserves the order of the sequence and can be used to calculate mean accuracy (mAcc) and success rate (SR) by aligning open-vocabulary customized plans with their closed-vocabulary counterparts and labels. The second mapping is overlap mapping. This procedure is identical to order mapping except that if $s_n$ maps to $p_m$, a follow up step $s_{n+1}$ can be mapped to any step $p_1, ..., p_M$ as long as $s_n$ and $s_{n+1}$ are distinct. For identical steps, $s_{n+1}$ cannot map to $p_m$. This type of mapping preserves an understanding of which steps in the ground-truth sequence are present in the predicted sequence, regardless of order. Thus, mIoU can be calculated from this mapping.

For each mapping type and dataset, there are between 15 and 20 human-created training examples that are provided to ChatGPT-4o as few-shot examples. We refer to these metrics as automatic SR, mAcc and mIoU (a-SR, a-mAcc, a-mIoU).

To assess the quality of customization, we use Few-Shot-In-Context Learning with GPT-4o to generate a "relevance score" that evaluates how well the plan incorporates input keywords. A rubric, scored from 1 to 5, measures this customization, rewarding plans that meaningfully integrate the keywords and penalizing those that lack customization, regardless of overall planning success.

```
Rubric:
1:  The plan is not relevant to any of the keywords.
2:  The plan is somewhat relevant to a few keywords, but
lacks depth.
3:  The plan demonstrates a good balance of relevance,
either highly relevant to one keyword or moderately relevant
to all.
4:  The plan is relevant to most keywords, demonstrating a
strong application.
5:  The plan is highly relevant to all keywords, thoroughly
integrating them with clear and meaningful content.
```

Using this rubric, we create 20 examples each for CrossTask and COIN, providing them to the LLM in a few-shot learning setup.

**Aligned BERT Score (aBERT-Score).** To effectively measure the similarity between the predicted sequences and the generic plan, we further introduce a novel metric called aligned BERT Score. This metric is based on BERT similarity score (Zhang et al., 2020) and is calculated by applying an optimal alignment algorithm to both sequences, utilizing a similarity matrix $M[i][j]$. This matrix captures the cosine similarity between the embeddings of each action pair from the ground-truth sequence $a_{GT}^{\text{Generic}}$ and the customized predicted sequence $a_{Pred}^{\text{Customized}}$, as defined by the following equation:

$$M[i][j] = \text{CosineSimilarity}(a_{i,GT}^{\text{Generic}}, a_{j,Pred}^{\text{Customized}}) \quad (4)$$

In this equation, $a_{i,GT}^{\text{Generic}}$ denotes the $i^{th}$ reference action, while $a_{j,Pred}^{\text{Customized}}$ represents the $j^{th}$ hypothesis action. We then derive the similarity score associated with the trajectory corresponding to the optimal alignment path between the two sequences, which serves as a measure of their similarities. For further details on the workings of this metric, please refer to appendix B.

### 4.3 IMPLEMENTATION DETAILS

We implement the customized dataset using `GEMINI-1.5-Flash` as the VLM and `GPT-4o mini` as the LLM, which also serves as the judge for assessing customization loss during training (eq. (2)). To expand the training dataset, we generate pseudo-label customized plans for each sample by leveraging all combinations of sample `Keywords`, $\binom{K}{k}$, where $0 < k \leq K$, effectively increasing the dataset size.

We perform LoRA fine-tuning on the LlaVa-improved (13B parameter) (Liu et al., 2024a) model and full fine-tuning on Llava-Next (Liu et al., 2024b) , training each for three and four epochs to optimize performance on the validation set, respectively. Model evaluation is conducted on a held-out unseen test set. We use an initial learning rate of $2 \times 10^{-4}$ with a cosine learning rate scheduler and a training batch size of 16. The training process utilizes four NVIDIA A100 GPUs with 40GB of memory for LlaVa-improved and eight GPUs for Llava-Next. During training, we set $\lambda$ in eq. (3) to $5 \times 10^{-3}$.

### 4.4 COMPARISON WITH STATE-OF-THE-ART BASELINES

We assess CPP's performance in comparison to existing large models capable of customized procedure planning for instructional videos. Specifically, we use GPT-4o, a widely recognized and powerful vision-language model, as a baseline under zero-shot and few-shot regimes. Similar to CPP, GPT-4o is prompted with an initial visual observation and corresponding instructions for further details). We compare its performance to CPP models utilizing LlaVA-improved (i.e., LlaVA-1.5)

Figure 4: Example of CPP's output on two samples from CrossTask, showcasing the model's ability to generate plans conditioned on the visual state and input keywords.

Table 1: Comparison of CPP and state-of-the-art models on the CrossTask dataset. CPP demonstrates superior performance in planning and customization.

| models | a-SR↑ (%) | a-mAcc↑ (%) | a-mIoU↑ (%) | a-Relevance↑ (out of 5) | aBERT-Score↑ |
|---|---|---|---|---|---|
| GPT-4o mini (zero-shot) | 16.38 | 42.59 | 17.11 | 3.62 | 0.44 |
| GPT-4o mini (10-shot) | 19.22 | 45.81 | 20.89 | 3.74 | 0.50 |
| **CPP** (LlaVa-1.5 backbone with CL) | 30.75 | 62.06 | 48.55 | 3.72 | 0.60 |
| **CPP** (LlaVa-1.6 backbone with CL) | **32.30** | **64.13** | **50.65** | **3.89** | **0.67** |

and LlaVA-Next (i.e., LlaVA-1.6) backbones. To distinguish models trained with the customization loss introduced in eq. (2), we label them as "with CL" and "w/o CL" (CL referring to customization loss). The results presented in table 1 and table 2 highlight CPP's superiority across automatic metrics, including SR, Acc, mIoU, and aBERT-Score, for datasets CrossTask and Coin, demonstrating its advantages in planning, customization, and overall similarity to ground-truth plans.

Notably, CPP with the LlaVA-1.6 backbone outperforms GPT-4o's few-shot performance by 13.08%, 18.32%, and 29.76% in a-SR, a-mAcc, and a-mIoU, respectively, on the CrossTask dataset, and by 14.96%, 20.4%, and 28.87% on the COIN dataset.

In terms of customization, CPP performs competitively against GPT-4o, exceeding the a-Relevance score on CrossTask. GPT-4o's high score in this metric, however, results from over-customization of action steps based on the input keywords. GPT-4o leverages its vast prior knowledge to over-customize plans, adapting them beyond the natural levels found in instructional videos in an attempt to fully satisfy the input prompt.

Fig. 4 presents two sample predictions based on the given conditions and visual state for CPP (with the LlaVA-1.6 backbone and CL). As shown, the model accurately understands the initial task state and generates a plan that successfully meets the keyword conditions through to completion.

## 4.5 IMPACT OF CUSTOMIZATION LOSS

The integration of the customization loss into the overall objective function of the model (eq. (3)) significantly enhances CPP's performance, as illustrated in tables 3 and 4. In the CrossTask dataset, the a-Relevance score increases by 14 points, while the COIN dataset sees a rise of 25 points. Furthermore, this loss functions as a regularization mechanism, contributing to an overall improvement in planning scores, with a 1.26% increase in success rate for the COIN dataset. The tables also include the p-values for improvements in the a-Relevance score, highlighting the significance of these enhancements for each backbone.

Table 2: Comparison of CPP and state-of-the-art models on the COIN dataset. CPP demonstrates superior performance in planning and customization.

| models | a-SR↑ (%) | a-mAcc↑ (%) | a-mIoU↑ (%) | a-Relevance↑ (out of 5) | aBERT-Score↑ |
|---|---|---|---|---|---|
| GPT-4o mini (zero-shot) | 12.70 | 38.09 | 20.24 | 3.91 | 0.54 |
| GPT-4o mini (10-shot) | 16.50 | 41.18 | 23.07 | **4.11** | 0.58 |
| **CPP** (LlaVa-1.5 backbone with CL) | 29.37 | 58.92 | 49.70 | 3.82 | 0.65 |
| **CPP** (LlaVa-1.6 backbone with CL) | **31.46** | **61.58** | **51.94** | 4.04 | **0.72** |

Table 3: Impact of Customization Loss (CL) on the CrossTask dataset. The introduction of CL during training significantly enhances the model's performance across all metrics, particularly in a-Relevance.

| models | a-SR↑ (%) | a-mAcc↑ (%) | a-mIoU↑ (%) | a-Relevance↑ (out of 5) | aBERT-Score↑ |
|---|---|---|---|---|---|
| **CPP** (LlaVa-1.5 backbone w/o CL) | 30.17 | 61.88 | 48.16 | 3.58 | **0.61** |
| **CPP** (LlaVa-1.5 backbone with CL) | **30.75** | **62.06** | **48.55** | **3.72** Improvement p-value<.029 | 0.60 |
| **CPP** (LlaVa-1.6 backbone w/o CL) | 31.78 | 62.59 | 49.12 | 3.63 | 0.65 |
| **CPP** (LlaVa-1.6 backbone with CL) | **32.30** | **64.13** | **50.65** | **3.89** Improvement p-value<.047 | **0.67** |

Table 4: Impact of Customization Loss (CL) on the COIN dataset. The introduction of CL results in better customization.

| models | a-SR↑ (%) | a-mAcc↑ (%) | a-mIoU↑ (%) | a-Relevance↑ (out of 5) | aBERT-Score↑ |
|---|---|---|---|---|---|
| **CPP** (LlaVa-1.5 backbone w/o CL) | 29.24 | 58.42 | 50.55 | 3.75 | **0.66** |
| **CPP** (LlaVa-1.5 backbone with CL) | **29.37** | **58.92** | 49.70 | **3.82** Improvement p-value<.032 | 0.65 |
| **CPP** (LlaVa-1.6 backbone w/o CL) | 30.20 | 60.09 | 51.03 | 3.79 | 0.69 |
| **CPP** (LlaVa-1.6 backbone with CL) | **31.46** | **61.58** | **51.94** | **4.04** Improvement p-value<.018 | **0.72** |

## 4.6 CONCLUSION

In this study, we tackled the novel challenge of customized procedure planning in instructional videos by developing the Customized Procedure Planner (CPP) framework. Unlike previous approaches in CPPIV, which were limited to using only initial and final visual observations for procedure induction, CPP generates plans in a causal setting based on initial observations, along with user and task-specific requirements. CPP surpasses the state-of-the-art in existing models. A key innovation is the use of weak supervision by customizing existing PPIV datasets. This is achieved by extracting video-specific customization information from video samples and utilizing advanced LLMs. Our model also incorporates a novel LLM-based objective function during training to further enhance customization. We evaluate CPP using new metrics designed specifically for this setting, demonstrating its superiority in CPPIV. Looking ahead, we see potential for applying CPP to more diverse scenarios and generating customized plans for unseen tasks. Additionally, developing a high-quality customized dataset will pave the way for more advanced applications in this field.

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
