## A FURTHER ABLATION STUDY

**Impact of Customization Loss Factor.** To investigate the impact of the customization loss factor $\lambda$ as defined in eq. (3), we conducted experiments with various values of $\lambda$. The results of this experimentation are detailed in table A for the CrossTask dataset. We find $\lambda = 5e - 3$ to be optimal.

Table A: Ablation study on the effect of $\lambda$ on the a-Relevance score for CrossTask, using the CPP with the LlaVa-1.6 backbone.

| $\lambda$ | 1e-2 | 5e-3 | 1e-3 |
|---|---|---|---|
| a-Relevance | 3.71 | **3.89** | 3.80 |

**Ablation Study on the Quality of Pseudo-Labels.** To evaluate the quality of the pseudo-labels, we conduct a manual assessment for both the COIN and CrossTask datasets. A subset of 50 samples is selected from each dataset, and we pose three targeted questions to assess the planning and customization attributes of the generated plans: 1. How well does the customized procedure plan align with the instructions provided in the video? 2. How effectively does the customized procedure plan achieve the end state depicted in the video? and 3. How well-customized is the procedure plan to the specific task example shown in the video? we rate each plan on a scale of 1 to 5, with ratings being: 1- Not accurate/effective/customized at all, 2- Somewhat accurate/effective/customized, 3- Neutral, 4- Somewhat accurate/effective/customized, 5- Very accurate/effective/customized. The table B shows the result for this assessment.

To further assess how well the customization process preserves the integrity of the plans (i.e., action order), we measure the pseudo-labels' SR, mAcc, and mIoU, along with the aBERT-Score, in comparison to the generic ground truth plans. table C presents the results of this comparison, highlighting the extent to which the essence of the plans is maintained during the customization process.

Table B: Ablation study results on the quality of pseudo-labels. Q1) Alignment of the plan with the instructional video, Q2) Effectiveness of the plan in achieving the end state, and Q3) Customization level of the plan to the video content.

| Questions | Q1 ↑ | Q2 ↑ | Q3 ↑ |
|---|---|---|---|
| CrossTask | 3.80 ±0.14 | 3.53 ±0.17 | 4.18 ±0.14 |
| COIN | 3.63 ±0.22 | 3.47 ±0.22 | 3.80 ±0.22 |

Table C: Ablation study on the similarity between pseudo-labels and ground-truth generic plans for the CrossTask and COIN datasets.

| models | a-SR↑ (%) | a-mAcc↑ (%) | a-mIoU↑ | aBERT-Score↑ |
|---|---|---|---|---|
| CrossTask | 89.37 | 95.60 | 97.41 | 0.79 |
| Coin | 87.77 | 95.34 | 96.28 | 0.82 |

## B ALIGNED BERT SCORE: OPTIMAL PATH ALIGNMENT

In this section, we detail the process of finding the optimal alignment path between the predicted and reference action sequences using a dynamic programming approach to measure the aligned BERT score. We define $\text{Score}(i, j)$ as the optimal score up to the $i^{th}$ reference action and the $j^{th}$ hypothesis action. The recursive formula for calculating this alignment considers three potential scenarios:

$$\text{Score}(i,j) = \max \begin{cases} \text{Score}(i-1, j-1) + M[i][j] & \text{(match)} \\ \text{Score}(i-1, j) - \delta & \text{(gap in hypothesis)} \\ \text{Score}(i, j-1) - \delta & \text{(gap in reference)} \end{cases}$$

In this formula, $M[i][j]$ represents the cosine similarity between the $i^{th}$ reference action and the $j^{th}$ predicted action, while $\delta$ denotes the penalty for introducing gaps into either sequence.

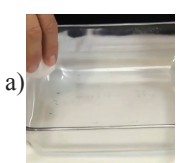

a)

Task: Make French Toast

Keywords: Peanut Butter, Banana, Sandwich

Prediction:
- take the French toast sandwich from the pan,
- spread peanut butter on the toast and slice bananas to top it.

GT generic plan:
pour egg, whisk mixture, pour milk, add vanilla extract, whisk mixture, dip bread in mixture, put bread in pan, flip bread, remove bread from pan, top toast

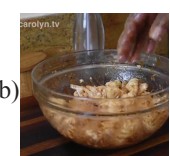

b)

Task: Make Taco Salad

Keywords: Black Bean, Cauliflower

Prediction:
- mix in cauliflower as a meat substitute,
- stir in black beans and diced onions,
- and combine all ingredients for an even texture.

GT generic plan: stir, add onion, stir, stir, stir

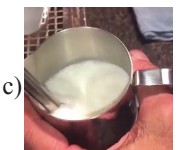

c)

Task: Make a Latte

Keywords: Espresso, Milk Texturing, Latte Art, Morning

Prediction:
- Steam milk until velvety,
- pour into a cup,
- add a shot of espresso and texture the milk.
- Pour more milk for layering
- top with frothy milk,
- create latte art, and finish with a drizzle of milk

GT generic plan:
steam milk, pour milk, pour espresso, steam milk, pour milk, steam milk, pour milk, pour milk

Figure 5: Examples of failure cases encountered by the model, Each scenario highlights specific challenges faced by the model in understanding visual states and incorporating relevant keywords.

After computing the optimal alignment score, denoted as score, we normalize it based on the lengths of the ground-truth sequence $L_{GT}$ and the predicted sequence $L_{pred}$. The normalized score, referred to as scaled_score, is calculated using the following formula:

$$\text{scaled\_score} = \frac{\text{score}}{\sqrt{(L_{GT} + 1)(L_{pred} + 1)}}$$

This normalization ensures that the score remains proportional to the lengths of both sequences, allowing for a fair comparison across varying sequence lengths. Ultimately, this method effectively captures the nuanced correspondence of actions, providing a comprehensive evaluation of model performance in aligning predicted sequences with ground-truth actions.

## C FAILURE CASE STUDY

Fig. 5 presents instances where the model faced challenges, leading to discrepancies in the predicted sequences. These examples highlight potential reasons for the model's divergence in the prediction:

- In scenario a), the model struggles to understand the visual state of the image, misinterpreting it as laying the already-made French toast in the dish rather than breaking the egg as the initial action. This results in miss-planning and neglecting some keywords. - In scenario b), the model generates a reasonable plan, but the order of actions differs from that of the ground truth generic plan. - Finally, in scenario c), the model fails to incorporate one of the keywords.