# OpenReview forum: "Customized Procedure Planning in Instructional Videos"
_ICLR.cc/2025/Conference — Submitted to ICLR 2025_

### Official Review · Reviewer_Pgdi · 2024-10-27

**Soundness:** 3
**Presentation:** 3
**Contribution:** 2
**Rating:** 5
**Confidence:** 4

**Summary:**

This paper investigates a more practical formulation of PPIV that considers user directions, called customized procedure planning in instructional videos.To overcome data limitations, the authors built a novel pipeline to collect customizations from existing PPIV datasets. Finally, a Customized Procedure Planner (CPP) framework (based on Llava) with a customization loss is proposed.

**Strengths:**

1.  The article investigate an interesting problem, extend conventional procedure planning to open-vocabulary, varied, and detailed instructional plans.
2. The authors propose a weakly supervised training approach that addresses the lack of customization annotations for CPPIV model training.

**Weaknesses:**

1.  This paper builds a pipeline for generating Customized Plans. The authors need to provide more examples and statistical results to demonstrate the effectiveness of the generated plans. Additionally, most of the keywords provided as examples in the manuscript are materials used in the production process, which does not quite align with the notion of customization.
2. In the data collection pipeline, does the VLM input include a Generic Plan? The prompt in Figure 2 does not seem to contain this information. Does the VLM model have such capabilities or knowledge?
3. In this task, the ground truth (GT) usually consists of gerund phrases, while the model generates full sentences. Could adding related prompts to constrain the model's output improve performance?
4. The authors should provide more examples to demonstrate the effectiveness of the proposed model and modules outlined in the text, Instead of just using numerical results.
5. The impact of the Customization Loss is marginal.

**Questions:**

Please refer to weaknesses.

---

> ### Author Response · Authors · 2024-11-24
>
> > Response to point 1:
>
> Thank you for your feedback. We appreciate your suggestions and will include more example outputs in the camera-ready version to better demonstrate the effectiveness of the generated plans.
>
> Regarding your point on customization keywords, we believe one of the strengths of our work lies in how we collect these keywords. Rather than predefining keywords or constraints, we directly extract customization keywords from a large-scale set of real-life YouTube instructional videos. Some videos are more generic, while others are highly specific and customized, reflecting the diversity of real-world content. This approach avoids introducing customization bias, as it reflects the natural variability in how users express their needs, rather than simulating customized keywords.
>
> By processing and extracting keywords directly from the dynamic content of real-world videos, our model generates more authentic and user-aligned plans. This method is less imposed and more reflective of actual user behavior, where customization evolves from the content itself, ensuring that the generated plans are truly tailored to the task at hand.

---

> > ### Author Response · Authors · 2024-11-24
> >
> > > Response to point 2:
> >
> > Thank you for your question. Yes, in the data collection pipeline, the VLM (Vision-Language Model) does indeed take the generic plan as part of the input text. For the sake of clarity and space in the figure, we did not include this information in the shortened prompt shown in Figure 2. However, the actual prompt (Prompt 2) does include the generic steps. We will make this point clearer in the camera-ready version for further clarity.

---

> > > ### Author Response · Authors · 2024-11-24
> > >
> > > > response to point 3:
> > >
> > > Thank you for your comment. There exists a trade-off. While constraining the model’s outputs to gerund phrases could improve alignment with ground truth for evaluation, it would also limit the model’s ability to generate detailed, customized plans.
> > >
> > > We intentionally did not impose this restriction, as we aimed to preserve the open-vocabulary quality of our model and allow for diverse, detailed responses tailored to user needs. Our experiments indicated that limiting outputs to phrases improved alignment but resulted in worse customization. Therefore, we prioritized maintaining the flexibility and adaptability of the model to generate more nuanced, user-specific plans.

---

> > > > ### Author Response · Authors · 2024-11-24
> > > >
> > > > > Response to point 4:
> > > >
> > > > Thank you for your feedback. We appreciate your suggestion and will include additional examples in the camera-ready version to better demonstrate the effectiveness of our proposed model and the modules outlined in the text, to ensure a more comprehensive understanding of our model's performance alongside the numerical results.

---

> > > > > ### Author Response · Authors · 2024-11-24
> > > > >
> > > > > > Response to point 5:
> > > > >
> > > > > Thank you for your comment. While the improvements may appear small, they are statistically significant, as demonstrated by the significance tests in Tables 3 and 4. We believe this shows that the customization loss acts as a regularizer, leading to a meaningful improvement in the model's performance.

---

> > > > > > ### Author Response · Authors · 2024-11-25
> > > > > >
> > > > > > Dear Reviewer,
> > > > > >
> > > > > > Thank you for your feedback. We’ve addressed your comments in our response on OpenReview. As the discussion phase ends on November 26, we’d appreciate it if you could confirm if your concerns are resolved and consider updating your scores.
> > > > > >
> > > > > > Thank you!

---

> > > > > > > ### Comment · Reviewer_Pgdi · 2024-11-27
> > > > > > > **Response to the authors**
> > > > > > >
> > > > > > > Thanks for your response. My concerns are partially addressed. I decide to keep my score, as the paper still some distance from the acceptance threshold and needs further refinement.

---

### Official Review · Reviewer_m1et · 2024-11-02

**Soundness:** 2
**Presentation:** 3
**Contribution:** 2
**Rating:** 5
**Confidence:** 5

**Summary:**

To address the potential challenges of lacking details in action steps in procedure planning in videos, the authors propose Customized Procedure Planner (CPP) framework to predict detailed action steps. They also used foundation models to create detailed action labels for benchmark dataset COIN and CrossTask. They also propose automated LLM-based metrics to evaluate the proposed models, therefore setting baselines.

**Strengths:**

-The authors pinpoint the problem of lacking detailed action steps in textual form that could distinguish the completion of task in procedure planning.

-To address this problem, they proposed the CPP framework and leveraged foundation models to train the model with pseudo action labels. The experiments in work is extensive.

**Weaknesses:**

-The authors failed to advocate the gravity and scientific significance of the problem (e.g. lack of detailed action steps or user requirements) that they were trying to solve.

-It seems that both are achievable just expanding the input/output spaces of previous tasks.

-The proposed framework lacks novelty. It is a combination of foundation models, designed to solve a very specific task.

-The authors only compare their proposed model with two foundation model baselines.

-Using foundation models to do the evaluation are not robust because the definition of task success is not based on ground-truth.

**Questions:**

-How is user requirements and detailed action steps fundamentally different from previous task instruction and action outputs? Can I view them as merely adding more detail to the data in the same modality?

-What solving this task matter? Can you prove or is there evidence that it might have impact other fields (e.g. in real world robotics?)

-Have you tried other combination of foundation models to solve your task?

-How should you reproduce your experiment results since commercial foundation model outputs are not reproducible?

-Why choose COIN and CrossTask datasets? There are more recent datasets (e.g. Ego4D).

---

> ### Author Response · Authors · 2024-11-24
>
> > Response to first comment
>
> Thank you for your feedback. We respectfully disagree and believe the manuscript adequately addresses the gravity and scientific significance of the problem.
>
> 1. **Introduction (Lines 046–079)**:
>    We highlight the *semantic gap* in generating detailed and customized plans—a key limitation in prior literature—underscoring the need for user-specific, tailored plans rather than generic task objectives.
>
> 2. **Figure 1 Illustration**:
>    Figure 1 contrasts our approach with prior methods, which generate generic action labels. Our framework produces detailed, customized plans that incorporate user-specific needs and task conditions.
>
> 3. **Lines 086–093**:
>    We explain the need for models to integrate both the current visual state and user-specific textual requirements, enabling tailored, user-aligned plans. Figure 1b further emphasizes the practical value of this approach in comparison to traditional models.
>
> 4. **Lines 094–099**:
>    We discuss how closed-vocabulary settings restrict predictions to predefined labels, leaving the challenge of generating detailed, user-specific plans unresolved.
>
> 5. **Lines 099–107**:
>    We address the significant gap in datasets for training customized procedure planners. Existing datasets, such as CrossTask and COIN, lack detailed, time-stamped plans tailored to specific videos and user requirements. This limitation is further explored in Section 3.3.
>
> 6. **Lines 123–131**:
>    We introduce a novel evaluation framework, addressing the inadequacy of traditional metrics for open-vocabulary, customized plans. We propose LLM-based metrics and validate our model’s performance on CrossTask and COIN datasets, establishing a strong baseline that outperforms state-of-the-art approaches.
>
> These points collectively advocate the significance of the problem and highlight our contributions in addressing these challenges. However, we appreciate your feedback and would welcome specific suggestions on areas you feel were overlooked so we can further improve the manuscript.

---

> > ### Author Response · Authors · 2024-11-24
> >
> > > Response to the second comment
> >
> > Thank you for your comment. We respectfully disagree, as expanding the input/output spaces alone captures only one aspect of the solution. Our setting fundamentally differs from prior tasks in several key ways:
> >
> > 1. **Dual Goal Representation**:
> >    Traditional procedure planning approaches are primarily goal-oriented, typically defining the goal using a single image. In contrast, our approach represents the goal through an *Objective* and specific *Conditions*, creating a more nuanced and customizable representation tailored to user requirements.
> >
> > 2. **Beyond Generic Transitions**:
> >    Prior tasks focus on decoding generic transitions between two visual states. Our approach significantly expands on this by introducing a setting that addresses a dual goal-oriented problem: satisfying both the task-specific visual objectives and user-defined directions.
> >
> > 3. **Integration of Visual and Textual Inputs**:
> >    Our framework requires a model to process visual content, understand the current state, identify involved objects, comprehend task objectives, and align these with user-specified directions. This combination results in a more practical and comprehensive approach, addressing real-world applications where customization and adaptability are essential.
> >
> > 4. **New Training and Evaluation Framework**:
> >    Training and evaluating such models require a novel framework not addressed in prior tasks. The outputs of these models expand beyond classic metrics like SR, ACC, and IoU by incorporating subjectivity and user customization as added dimensions. This introduces unique challenges, as models must perform not only on structural accuracy but also on alignment with user-specific needs and conditions. Our proposed evaluation framework addresses these new dimensions, further emphasizing the distinction of our setting from traditional approaches.
> >
> > This fundamental shift highlights the limitations of prior methods and the advancements introduced in our work, providing a more practical and effective solution for customized procedure planning.

---

> > > ### Author Response · Authors · 2024-11-24
> > >
> > > > Response to comment 3
> > >
> > > Thank you for your comment. To clarify, we would like to summarize the main contributions of our paper:
> > >
> > > 1. **Novel Setting of Customized Procedure Planning**:
> > >    We introduce a more practical formulation of procedure planning for instructional videos, shifting the focus to generating instructional plans that cater to user-specific needs rather than generic task completion. This novel setting emphasizes incorporating user directions and requirements into task planning.
> > >
> > > 2. **Customized Procedure Planner Framework**:
> > >    Our proposed CPP framework generates open-vocabulary instructional plans tailored to user-specified condition keywords, enabling the transformation of initial visual states into task completion in a way that aligns with user goals.
> > >
> > > 3. **Weakly Supervised Training Approach**:
> > >    We address the challenge of limited customization annotations by proposing a weakly supervised training approach. This allows the CPP framework to learn customized planning from unannotated videos, overcoming a key barrier in model training for this setting.
> > >
> > > 4. **Extended Evaluation Metrics**:
> > >    We extend conventional procedure planning metrics to assess open-vocabulary, varied, and detailed instructional plans. This comprehensive evaluation framework measures both planning and customization performance, capturing dimensions previously overlooked in the field.
> > >
> > > We hope this clarification highlights the significance and originality of our work.

---

> > > > ### Author Response · Authors · 2024-11-24
> > > >
> > > > > Response to comment 4:
> > > >
> > > > Thank you for your feedback. We appreciate your concern regarding the choice of baselines. For our evaluations, we intentionally selected the strongest foundation model baselines to assess the task at hand. Specifically, we compared our proposed model with:
> > > >
> > > > 1. **GPT-4o**: A state-of-the-art model widely regarded for its performance and robustness in generating text and understanding complex inputs.
> > > > 2. **LLaVA 1.5 and LLaVA-Next**: Two state-of-the-art open-source models specifically designed for vision-language tasks, ensuring a strong and relevant comparison for our customized procedure planning framework.
> > > >
> > > > By evaluating against these leading models, we ensured a rigorous assessment of our approach. We hope this clarification addresses your concern.

---

> > > > > ### Author Response · Authors · 2024-11-24
> > > > >
> > > > > > Response to comment 5:
> > > > >
> > > > > Thank you for your comment. We respectfully disagree and would like to clarify that our evaluation metrics are indeed based on ground-truth generic plans, as detailed in Section 4.2, *Automatic Metrics*. This study evaluates performance along two critical dimensions: **planning quality** and **customization quality**.
> > > > >
> > > > > 1. **Ground-Truth Alignment**:
> > > > >    For planning quality, our approach aligns open-vocabulary customized plans with closed-vocabulary generic ground-truth sequences. This alignment is performed using two robust sequence mappings:
> > > > >
> > > > >    - **Order Mapping**:
> > > > >      This sequential process maps each ground-truth step to a corresponding predicted step while preserving the sequence order. Steps without valid matches are marked as missing, enabling the calculation of metrics such as **mean accuracy (mAcc)** and **success rate (SR)**.
> > > > >
> > > > >    - **Overlap Mapping**:
> > > > >      Similar to order mapping but less constrained, this method identifies the presence of ground-truth steps in the predicted sequence regardless of order. This mapping supports the calculation of **mean Intersection over Union (mIoU)**, providing additional insights into sequence coverage.
> > > > >
> > > > > 2. **Few-Shot In-Context Learning**:
> > > > >    To achieve accurate mappings, we use a a systematic approach and use Few-Shot In-Context Learning with GPT-4o, providing 20 human-created training examples per dataset. These examples serve as benchmarks, ensuring robust and consistent evaluations of the predicted plans against ground-truth sequences.
> > > > >
> > > > > 3. **Novel Metrics for Open-Vocabulary Frameworks**:
> > > > >    Given the nature of our open-vocabulary framework, traditional closed-vocabulary metrics are insufficient. Our proposed automatic metrics (a-SR, a-mAcc, a-mIoU) extend standard planning metrics to robustly quantify performance in this more complex setting, ensuring both planning quality and alignment with ground truth.
> > > > >
> > > > > We hope this explanation clarifies that our evaluation framework is grounded in rigorous alignment with ground-truth plans and combines robust methodologies to address the complexities of open-vocabulary procedure planning.

---

> > > > > > ### Author Response · Authors · 2024-11-24
> > > > > >
> > > > > > > Response to Question 1:
> > > > > >
> > > > > > No, user requirements and detailed action steps are not merely additional detail within the same modality. While the added detail is a benefit of our framework, the main focus is on defining the task objectives and conditions. The output must not only be more detailed, but also bring about a state transition that aligns with both the conditions and objectives specified by various modalities and user directions. This requires the model to integrate visual input and user-specific requirements, resulting in a more dynamic and customized approach to task execution.
> > > > > > > Response to Question 2:
> > > > > >
> > > > > > Thank you for your question. Solving this task is important because it addresses the gap in generating **customized, user-specific procedure plans**—a capability that has far-reaching implications across multiple fields.
> > > > > >
> > > > > > 1. **Impact on Real-World Robotics**:
> > > > > >    In robotics, tasks often require high levels of customization to adapt to varying user needs, environmental conditions, and specific goals. Our framework's ability to incorporate user-specific directions and handle complex, open-vocabulary objectives is directly applicable to **personalized robotics**, where tasks must be performed based on individual preferences or dynamic environments. For example, in assistive robotics, customizing task instructions (e.g., for elderly care or rehabilitation) based on user needs would significantly improve task effectiveness and user satisfaction.
> > > > > >
> > > > > > > Response to Question 3:
> > > > > >
> > > > > > 2. **Broader Applications**:
> > > > > >    Beyond robotics, our approach can impact fields like **adaptive learning systems**, **human-computer interaction**, and **personalized training**. The ability to generate task plans that are tailored to specific user requirements could enhance systems that require personalized instructions, such as educational platforms, smart assistants, or industrial automation tools.
> > > > > >
> > > > > > 3. **Evidence of Impact**:
> > > > > >    While this work is foundational, there is growing interest in applying customized procedure planning in areas like **autonomous systems** and **personalized AI assistants**, where context and user specifications are critical. We envision that future studies and applications of this model will demonstrate its utility in real-world scenarios, as seen with the growing adoption of personalized AI in health, education, and assistive technologies.
> > > > > >
> > > > > > Thus, solving this task addresses a core need in many practical domains, paving the way for more **adaptive, user-centric systems** with significant real-world impact.
> > > > > > > Response to Question 4:
> > > > > >
> > > > > > Thank you for your question. We primarily experimented with LLaVA, LLaVA Next, and GPT-4o for our task. The main focus of our work was on the novel setting and supervision method, as well as leveraging the capabilities of these models to handle user-specific customization. While we recognize that other foundation models could be explored, our primary contribution lies in the unique task formulation and the novel approach to weak supervision, which we believe distinguishes this work.
> > > > > > > Response to Question 5:
> > > > > >
> > > > > > Thank you for your comment. We specifically chose LLaVA and LLaVA Next for this problem due to their open-source nature, which ensures reproducibility. While we have introduced a systematic framework for maximizing reproducibility with commercial AI models, we acknowledge that exact reproduction of results with commercial foundation models remains an ongoing challenge in the AI field. Addressing this issue is critical and continues to be an area of focus for future research in the broader AI community.
> > > > > > > Response to Question 6:
> > > > > >
> > > > > > Thank you for your comment. We chose the COIN and CrossTask datasets primarily because they are widely used in the community for evaluating multi-modal procedure planning, offering a well-established benchmark for this specific task. These datasets provide detailed annotations that are essential for evaluating the performance of customized procedure planning models in instructional video contexts.
> > > > > >
> > > > > > While newer datasets like Ego4D are available and offer valuable insights, it’s important to note that **Ego4D** is focused on egocentric problems, which is a different problem and domain from instructional procedure planning. Ego4D is a controlled dataset designed for egocentric task understanding, and it often involves much longer action sequences that may include irrelevant actions not directly related to the main task. This makes it less suitable for evaluating the specific goals of instructional procedure planning, which requires detailed, task-specific instructions for generating customized plans.
> > > > > >
> > > > > > Given the focus of our work, we selected COIN and CrossTask because they better align with the goals of our research. However, we plan to extend our experiments to newer datasets, including Ego4D, in future work to evaluate the scalability and robustness of our approach across different domains.

---

> > > > > > > ### Author Response · Authors · 2024-11-25
> > > > > > >
> > > > > > > Dear Reviewer,
> > > > > > >
> > > > > > > Thank you for your feedback. We’ve addressed your comments in our response on OpenReview. As the discussion phase ends on November 26, we’d appreciate it if you could confirm if your concerns are resolved and consider updating your scores.
> > > > > > >
> > > > > > > Thank you!

---

> > ### Comment · Reviewer_m1et · 2024-11-26
> >
> > Thank you for your response. My concerns are partially addressed.

---

### Official Review · Reviewer_UHFc · 2024-11-02

**Soundness:** 3
**Presentation:** 2
**Contribution:** 2
**Rating:** 6
**Confidence:** 4

**Summary:**

The paper introduce a new task called customized procedure planning (CCP) as an extension to the task of procedure planning in instructional videos. This task generates action plans in natural languages conditioned on user-specific requirements and task objectives, utilizing a weakly supervised approach to overcome the lack of detailed customization annotations in existing datasets. The authors propose a training method, leveraging Large Language Models (LLMs) like GPT-4 for generating pseudo-labels and for enhancing customization through a novel objective function. The paper also introduces new LLM-based metrics to evaluate open-vocabulary, user-specific plans.

**Strengths:**

1. The introduction of CPPIV is a valuable extension of traditional procedure planning tasks, addressing the limitations of existing models that do not output natural language plan.

2. The development of LLM-based metrics to evaluate open-vocabulary plan customization and quality is innovative.

**Weaknesses:**

1. While the new metrics are interesting, the reliance on LLM-based evaluation could be perceived as less interpretable and overly dependent on the LLM’s performance and biases.

2. Rather than simply framing this task as catering to user-specific needs, the primary distinction lies in how the goal is represented. Traditional procedure planning approaches are goal-oriented, often defining the goal using a single image. In contrast, this approach defines the goal using an Objective along with specific Conditions, providing a more nuanced and customizable representation.

3. The paper does not thoroughly address the potential limitations and biases introduced by pseudo-labeling, especially given that human-annotated datasets remain scarce.

4. The results on CrossTask and COIN are also somewhat difficult to interpret. Since these datasets lack ground truth action plans expressed in natural language, the evaluation relies on pseudo-labels generated by LLMs. This introduces a challenge: comparing model outputs, which are also generated by LLMs, against pseudo-labels from the same or similar models raises questions about the objectivity and robustness of the evaluation process.

**Questions:**

Could you elaborate on the potential biases introduced by using GPT-4 and GEMINI for pseudo-labeling and how they may affect the quality of the generated plans? How does the model handle cases where the user-specified conditions conflict with one another or with the task objective?

---

> ### Author Response · Authors · 2024-11-21
>
> Thank you for your thoughtful feedbacks. We provide the following responses to points made:
> > 1
>
> We agree that automatic metrics, including those based on LLMs, inherently carry some biases. However, we have taken several steps to minimize these biases and maximize accuracy:
>
> 1. **Coupling with Objective Metrics**:
>    Our LLM-based metrics (aSR, aAcc, aIoU, and aRelevance) are designed to complement each other and objective metrics like aBERT-Score. This balanced framework captures nuanced aspects of customization through LLMs while maintaining interpretability via objective, quantitative measures.
>
> 2. **Systematic Design with Rubrics and Examples**:
>    To standardize LLM evaluations, we provide a detailed rubric and use few-shot in-context learning examples (4 examples per rubric rule) to anchor the model's outputs. This ensures that evaluations follow a consistent and systematic framework, reducing variability and improving reliability.
>
> 3. **Transparent Evaluation Process**:
>    To enhance transparency, for each evaluation instance, the LLM is asked to provide reasoning for order mapping, alignment to generic plans, and relevance before scoring based on the rubric. This systematic reasoning ensures logical consistency and provides traceability for the model’s ratings.
>
> 4. **Human Validation and Performance Consistency**:
>    Validation tests demonstrated high inter-rater agreement (>85%), confirming the reliability and consistency of our evaluation approach. By combining LLM-based evaluations with human validation and objective metrics, we minimize potential biases while maximizing the robustness of our framework.
>
> We believe this combined approach provides a reliable and interpretable evaluation methodology while addressing the inherent challenges of automatic metrics.
>
> > 2
>
> Thank you for your insightful observation. We agree and believe that this distinction is indeed a strength of our framework. As correctly identified, defining the goal using an Objective along with specific Conditions enables a more nuanced and customizable approach compared to traditional goal-oriented methods.
>
> We will revise the introduction to bring greater attention to this key contribution.
>
> > 3 & 4
>
> We agree that potential bias is an inherent Challenge of using models like GPT-4o and GEMINI for generating pseudo-labels and weak-supervision frameworks. However, we have implemented several measures to minimize these biases and ensure the reliability of our pseudo-labels:
>
> 1. **Limited Reliance on Models:**
> While GPT-4o and GEMINI generate customization details, critical elements like plan actions and timestamps are manually annotated, reducing model bias.
>
> 2. **Rigorous Quality Assessment**:
> We manually evaluated the pseudo-labels on subsets of the COIN and CrossTask datasets (Section A), with high scores (e.g., 4.18/5 for customization on CrossTask), confirming label reliability.
>
> 3. **Independent Models**:
> GEMINI extracts customization keywords, and GPT-4o evaluates them, ensuring independence. Both models, trained on diverse datasets, reduce the risk of overlapping biases. We also use objective metrics like aBERT-Score (Table C) for unbiased model evaluation.
>
> 4. **Error Correction During Training**:
> GPT-4o is incorporated into the objective function to iteratively refine predictions, reducing the impact of pseudo-label inaccuracies.
>
> 5. **Acknowledgment of Limitations**:
> We recognize that biases are an inherent challenge in weak supervision. While pseudo-labels enhance model robustness, we will include a more detailed discussion of these limitations in the paper for transparency.
>
> > **Response to Reviewer Question:**
>
>
>
> 1. **Mismatch Between User Directions and Images**:
>    In our dataset, there is minimal mismatch between user-specified conditions and images, as both were sourced from aligned YouTube instructional videos. However, pseudo-labels generated by GPT-4o and GEMINI may still be biased toward task objectives, especially when the model prioritizes common patterns or assumptions from the training data.
>
> 2. **Task-Irrelevant Keyword Leakage**:
>    Despite careful dataset selection, pseudo-labels can be influenced by irrelevant information, such as frequently encountered task-related keywords or sequences. This bias may cause the model to over-prioritize certain objectives or generate labels based on less relevant data. Section C presents failure case studies that illustrate how these biases affect the model, particularly in edge cases or complex user conditions.

---

### Official Review · Reviewer_wx1o · 2024-11-09

**Soundness:** 3
**Presentation:** 2
**Contribution:** 2
**Rating:** 6
**Confidence:** 4

**Summary:**

The paper addresses the issue of generating customized procedures for task planning in instructional videos. Existing methods face challenges like overlooking customization and lacking proper datasets. The contributions are significant. It presents a novel setting for customized procedure planning, emphasizing user - specific needs. The Customized Procedure Planner (CPP) framework is proposed, which utilizes LlaVa - based models and is trained with pseudo - labels generated through a weakly - supervised approach. New evaluation metrics are introduced to assess planning and customization quality. Experimental results on CrossTask and COIN datasets show CPP's superiority over baselines like GPT - 4o. The integration of customization loss further enhances performance. Overall, this research lays a strong foundation for future work in customized procedure planning.

**Strengths:**

1. The paper shows a novel task and CPP framework, using models creatively for generating customized procedures for task planning.

2.It is well-written and clear. The introduction motivates the problem, and the technical approach is detailed.

**Weaknesses:**

1. The CPP model is trained and evaluated on a specific set of instructional video tasks (mostly related to cooking and DIY activities in the used datasets). It is unclear how well the model would generalize to other types of tasks or domains that have different characteristics and action requirements.

2. The process of creating pseudo - labels using GPT - 4o and GEMINI might introduce some biases or inaccuracies.

3.The interpretation of the "relevance score" for customization quality assessment could be more straightforward. The rubric used to measure customization is somewhat subjective, and it might not be clear how different users would rate the relevance of a plan.

4.The human evaluation seems to focus mainly on validating the model's performance rather than exploring potential areas for improvement. A more in - depth qualitative analysis of the human feedback could uncover additional insights into the strengths and weaknesses of the CPP model and guide further refinements.

**Questions:**

See details in 'Weakness' section.

---

> ### Author Response · Authors · 2024-11-20
>
> > 1
>
> Thank you for raising the question of generalizability. CPP’s generalization can be evaluated across three dimensions:
>
> **Open-Vocabulary in Natural Language:**
> CPP operates in an open-vocabulary setting, enabling it to handle diverse natural language inputs and outputs, adapting flexibly to user-defined requirements (Tables [1, 2]).
>
> **Within-Domain Task Generalization:**
> Within defined domains, we evaluated CPP’s ability to perform on unseen tasks not encountered during training. As shown in the table below, CPP significantly outperforms baseline models on 8 unseen tasks from various domains within the COIN dataset (comprising 180 tasks), demonstrating its strong generalization capability to new tasks within familiar domains.
>
> | Models                        | a-SR↑ (%) | a-mAcc↑ (%) | a-mIoU↑ (%) | a-Relevance↑ | aBERT-Score↑ |
> |-------------------------------|-----------|-------------|-------------|--------------|--------------|
> | GPT-4o mini (10-shot)         | 16.50     | 41.18       | 23.07       | 4.11         | 0.58         |
> | CPP (LlaVa-1.6 backbone) on unseen tasks      | 22.11     | 48.91       | 34.62       | 3.84         | 0.64         |
>
> **Cross-Domain Generalization**:
> Generalizing across different domains (e.g., DIY tasks vs. computer graphics) is challenging due to structural data differences. While not the focus of this study, our experiments suggest that domain-specific adaptation, such as fine-tuning or few-shot learning, may be needed for optimal performance. Addressing these challenges is an exciting direction for future research.
> >2
>
> We agree that using GPT-4o and GEMINI to generate pseudo-labels may introduce biases. However, we have implemented several measures to minimize these biases and ensure label reliability:
>
> 1. **Limited Reliance on Models**:
>    While GPT-4o and GEMINI generate customization details, critical elements like plan actions and timestamps are manually annotated, reducing model bias.
>
> 2. **Rigorous Quality Assessment**:
>    We manually evaluated the pseudo-labels on subsets of the COIN and CrossTask datasets (Section A), with high scores (e.g., 4.18/5 for customization on CrossTask), confirming label reliability.
>
> 3. **Independent Models**:
>    GEMINI extracts customization keywords, and GPT-4o evaluates them, ensuring independence. Both models, trained on diverse datasets, reduce the risk of overlapping biases. We also use objective metrics like aBERT-Score (Table C) for unbiased model evaluation.
>
> 4. **Error Correction During Training**:
>    GPT-4o is incorporated into the objective function to iteratively refine predictions, reducing the impact of pseudo-label inaccuracies.
>
> 5. **Acknowledgment of Limitations**:
>    We recognize that biases are an inherent challenge in weak supervision. While pseudo-labels enhance model robustness, we will include a more detailed discussion of these limitations in the paper for transparency.
>
> >3
>
> Thank you for your feedback. We view the subjectivity of the relevance score as a strength, as customization inherently requires human judgment.
>
> 1. **Subjectivity by Design**:
>    Customization quality depends on how well a plan aligns with user-defined keywords, making subjectivity essential to capture nuanced aspects that automated metrics may miss. We formalized this with a structured rubric to ensure consistency while maintaining interpretive flexibility.
>
> 2. **Balanced Evaluation**:
>    To complement the relevance score, we use objective metrics like SR, a-mAcc, and aBERT-Score, which measure alignment, order, and similarity. This dual framework ensures a comprehensive evaluation, balancing human satisfaction with structural correctness.
>
> 3. **Scoring and Validation**:
>    The rubric includes clear guidelines and examples for each score level (1–5) and is reinforced with few-shot in-context learning (4 examples per rubric rule). Validation tests showed >85% inter-rater agreement, confirming its reliability.
>
> 4. **Dual Metrics Rationale**:
>    Combining subjective and objective metrics allows a holistic evaluation of customization quality—capturing both human satisfaction and the model’s alignment and accuracy.
>
> >4
>
> Thank you for your suggestion regarding deeper qualitative analysis of human feedback. Our current human evaluation focuses on validating the model's performance by assessing alignment, plan effectiveness, and customization quality using structured rubrics. While this approach demonstrates the model's effectiveness, we agree that a more detailed human evaluation would be valuable for uncovering improvement areas and is an interesting direction for future work.

---

### Author Response · Authors · 2024-11-24

We deeply appreciate the insightful feedback provided by the reviewers. We are pleased that Reviewer wx1o acknowledged the novelty and creativity of our CPP framework, highlighting its well-written problem definition, motivating introduction, and clear technical approach. Reviewer UHFc commended CPPIV for its contribution to extending traditional procedure planning and recognized the innovation of our LLM-based metrics for evaluating open-vocabulary plan customization. Reviewer m1et appreciated our identification of the gap in detailed action steps and the thoroughness of our experiments, while Reviewer Pgdi praised the significance of our weakly supervised training approach in addressing the challenge of lacking customization annotations. We are grateful for the recognition of the impact and importance of our work by all reviewers.

We have addressed their concerns  below, and hope they will update their scores if they find our responses clarifying.

---

### Meta-Review · Area_Chair_27ki · 2024-12-08

**Metareview:**

The paper works on a novel task and develops a Customized Procedure Planner (CPP) framework for generating customized procedures for task planning.

After rebuttal and discussion, this paper receives review scores of 5,5,6,6. So it is a paper on the fence.

Reviewers raised many concerns, such as lack of novelty, more solid results to demonstrate the effectiveness of the generated plans, more robust evaluation, and more comparisons. After the discussion, reviewers acknowledged that some of the concerns have been addressed, but many concerns remain, and the submission needs further refinement.

AC has checked the submission, the reviews, the rebuttal, and the discussions between the authors and the reviewers. AC appreciated the authors' effort in improving the submission in the discussion, as many new results were added. However, AC agreed with the reviewers that better evaluations of the generated plan need to be done beyond just using foundation models. For example, authors can evaluate the generated plan by executing it in closed-loop simulation, such as simulated robotic manipulation tasks. Also, AC echoed the comment from the reviewer that this submission lacks novelty and theoretical foundation in the sense that it uses a foundation model to achieve some specific tasks. There is no guarantee of the output. Thus, a rejection is recommended.

**Additional Comments On Reviewer Discussion:**

There were discussions between the authors and reviewers. Many concerns have been addressed. The reviewers have acknowledged the responses from the reviewers, two of them saying " My concerns are partially addressed.", and " My concerns are partially addressed. I decide to keep my score, as the paper still some distance from the acceptance threshold and needs further refinement.".

---

### Decision · Program_Chairs · 2025-01-22

Reject